

# Multiscale response of ionic systems to a spatially varying electric field

**Jesper S. Hansen**

"Glass and Time", IMFUFA, Department of Science and Environment, Roskilde University,
Postbox 260, DK-4000 Roskilde, Denmark

jschmidt@ruc.dk

## Abstract

In this paper the response of ionic systems subjected to a spatially varying electric field is studied. Following the Nernst-Planck equation, two forces driving the mass flux are present, namely, the concentration gradient and the electric potential gradient. The mass flux due to the concentration gradient is modelled through Fick's law, and a new constitutive relation for the mass flux due to the potential gradient is proposed. In the regime of low screening the response function due to the potential gradient is closely related to the ionic conductivity. In the large screening regime, on the other hand, the response function is governed by the charge-charge structure. Molecular dynamics simulations are conducted and the two wavevector dependent response functions are evaluated for models of a molten salt and an ionic liquid. In the low screening regime the response functions show same wavevector dependency, indicating that it is the same underlying physical processes that govern the response. In the screening regime the wavevector dependency is very different and, thus, the overall response is determined by different processes. This is in agreement with the observed failure of the Nernst-Einstein relation.



# 1   Introduction

Ionic liquids, molten salts and ionic solutions show a response to the application of an external electric field. In the case of small field amplitudes the response is typically modelled by linear constitutive relations, for example, the response manifested by a charge current is related to the local field by Ohm's law and the charge density to the external field through the charge-charge correlation function [1,2]. The response is characterized by different response functions (or transport coefficients) like the electric conductivity, electric permittivity and the charge-charge correlation. One can often find relations between the different response functions [2]; at least in some limits. A more famous one is the Nernst-Einstein equation that relates the self diffusion coefficient to the electric conductivity [3], i.e., single particle flux to the charge current. This is a quite surprising relation as the particle flux is a single particle phenomenon whereas the charge current is a collective phenomenon. The Nernst-Einstein equation is then only valid when the ion cross-correlations can be neglected [1,4]. One example where this assumption is not valid is where the flux of ion-pairs contributes to the mass flux, but not to the charge current as the charges cancels [2]. The deviation can be determined from simulations or experiments and is often quantified by a deviation parameter [2], which, interestingly, Harris et al. [5,6] have expressed in terms of the velocity cross-correlation functions. Importantly, the failure of the Nernst-Einstein equation means that the particle flux due to the electric field cannot be modelled through Ohm's law directly. Rather than approaching this problem through the deviation parameter it is appealing to take one step back and propose a linear constitutive relation that involves a new response function relating the mass flux to the external field directly. This is done in this paper.

The system's response is dependent on the wavelength of the external field, and this can be modelled through wavevector dependent response functions [2,7]. Investigating the wavevector dependence is relevant as the response can vary as function of length scale [8]. Also, this provide a way to probe a characteristic correlation lengths for a given system [8,9]; if the characteristic length scales are different for the different response functions this indicates that different physical underlying mechanisms are responsible for the system response. This is also addressed here.

The paper is organized as follows: In the next section the theory for the response of an ionic system subjected to a static sinusoidal external field is presented. In Sect. 3 molecular dynamics simulation results are presented and discussed, and, finally, in the last section conclusions from the work are drawn.

# 2   Theory

We consider an ionic system composed of one cation and one anion specie. The ions are rigid meaning that any higher order induced effects and electron transfer mechanisms are ignored. The charges are $\pm q$, respectively. Let $i$ indicate either a cation or an anion, i.e., $i = +$ or $-$, then the number density $n_i$ follows the balance equation [10]

$$\frac{\partial n_i}{\partial t} = \sigma_i - \boldsymbol{\nabla} \cdot n_i \mathbf{c}_i - \boldsymbol{\nabla} \cdot n_i \mathbf{u}, \tag{1}$$

where $n_i \mathbf{c}_i$ is the diffusive flux and $n_i \mathbf{u}$ the advective flux. The production term $\sigma_i$ accounts for additional forces that generate a local change in $n_i$; this includes application of an external electric field. The terms on the right-hand side of Eq. (1) can be expressed as the divergence of fluxes such that if one writes the production term as $\sigma_i = -\boldsymbol{\nabla} \cdot \mathbf{j}_i^e$ and $n_i \mathbf{c}_i = \mathbf{j}_i^d$ we have for

zero advection

$$\frac{\partial n_i}{\partial t} = -\nabla \cdot \mathbf{j}_i = -\nabla \cdot (\mathbf{j}_i^e + \mathbf{j}_i^d). \tag{2}$$

The system is kept away from equilibrium by application of a static spatially varying external electric field. The field points and varies in the direction parallel to the system $z$-direction, i.e, the non-zero $z$-component of the external field reads

$$E_z^{\text{ext}}(z) = E_0 k_n^m \cos(k_n z), \tag{3}$$

where $k_n = 2\pi n / L_z$ is the wavevector, $n = 1, 2, \ldots$, and $L_z$ is the length of the system in the $z$-direction. $m$ is either 0 or 1. The experimental realization of this field is not straightforward. Here it is considered as we are interested in the wavevector dependent response and as such this resembles the sinusoidal transverse and longitudinal force field methods (STF and SLF), see for example Refs. [11–13]. The corresponding electric potential is

$$\phi^{\text{ext}}(z) = -\int_0^z E_z^{\text{ext}}(z') \, dz' + \phi^{\text{ext}}(0) = -E_0 k_n^{m-1} \sin(k_n z), \tag{4}$$

using $\phi^{\text{ext}}(0) = 0$. Note, $m = 0$ corresponds to a wavevector independent field amplitude and $m = 1$ to wavevector independent potential amplitude.

It is in place to discuss the Maxwell equations. First, the induced (screening) field is $E = E(z)$ and according to Gauss' law $dE/dz = \rho_q/\epsilon_0$, where $\rho_q$ is the charge density given by the induced ionic density, $\epsilon_0$ is the electric permittivity of free space. From the Maxwell-Faraday equation $\nabla \times \mathbf{E} = -\dot{\mathbf{B}} = \mathbf{0}$, that is, the field due to the screeing does not result in any change in the magnetic field $\mathbf{B}$. Then Gauss' law for the magnetic field is fulfilled, $\nabla \cdot \mathbf{B} = 0$. Furthermore, since $\nabla \times \mathbf{B} = \mathbf{0}$ and $\dot{\mathbf{E}} = \mathbf{0}$ there are no net charge current (Ampere's circuital law). The system is therefore in a steady state.

To proceed one needs to relate the fluxes with the corresponding forces [10]. For sufficiently small force amplitude this is done through the generalized linear response theory. Consider the mass flux in the $z$-direction $j_i$ to depend on $N$ forces $X_n$, $n = 1, 2, \ldots N$ then we have in the homogeneous situation

$$j_i = -\sum_n \int_0^\infty \int_{-\infty}^\infty \chi_n'(\mathbf{r} - \mathbf{r}', t - t') X_n(\mathbf{r}', t') \, d\mathbf{r}' dt', \tag{5}$$

where $\chi_n'$ is the response function relating the flux $j_i$ to the force $X_n$. Since the system is in a steady state we can safely ignore time memory effects and, furthermore, assuming isotropy the response functions can then be written as $\chi_n'(\mathbf{r} - \mathbf{r}', t - t') = \chi_n(z - z')\delta(t - t')$. The flux is

$$j_i = -\sum_n \int_0^\infty \delta(t - t') \int_{-\infty}^\infty \chi_n(z - z') X_n(z', t') \, dz' dt'$$
$$= -\sum_n \int_{-\infty}^\infty \chi_n(z - z') X_n(z', t) \, dz' = -\sum_n \int_{-\infty}^\infty \chi_n(z - z') X_n(z') \, dz'. \tag{6}$$

The final expression is due to the steady state conditon. This generalized response formalism can be applied to the present situation. The flux is proposed to be given by the two terms ($N = 2$)

$$j_i = j_i^d + j_i^e = -\int_{-\infty}^\infty D_i(z - z') \frac{dn_i}{dz'} \, dz' - \frac{1}{q_i} \int_{-\infty}^\infty \chi_i(z - z') \frac{d\phi^{\text{ext}}}{dz'} \, dz', \tag{7}$$

where $D_i$ is the diffusion response function (or diffusion coefficient) and $\chi_i$ is the response function that relates the *mass* flux to the external field. The first relation is simply a generalized version of Fick's law, but the second relation is not a generalization of Ohm's law as $\chi_i$

relates the mass flux directly to external electric potential. Also, the force in Ohm's law is given by the *local* electric potential, i.e., the sum of the induced potential and the external potential. The $\chi_i$-response function can be interpreted as the system response to an external field excluding the effects from diffusion. Note that Eq. (7) is a generalized form of the Nernst-Planck equation [14].

In the steady state $j_i^d + j_i^e = 0$, and one has

$$\int_{-\infty}^{\infty} D_i(z-z')\frac{dn_i}{dz'}\,dz' = -\frac{1}{q_i}\int_{-\infty}^{\infty} \chi_i(z-z')\frac{d\phi^{\text{ext}}}{dz'}\,dz'. \tag{8}$$

In Fourier space by the convolution theorem for wavevector $\mathbf{k} = (0,0,k_n)$ this reads

$$ik_n\widetilde{D}_i(k_n)\widetilde{n}_i(k_n) = -\frac{ik_n}{q_i}\widetilde{\chi}_i(k_n)\widetilde{\phi}^{\text{ext}}(k_n), \tag{9}$$

or

$$\widetilde{n}_i(k_n) = -\frac{\widetilde{\chi}_i(k_n)}{q_i\widetilde{D}_i(k_n)}\widetilde{\phi}^{\text{ext}}(k_n). \tag{10}$$

Equation (10) is the expression for the Fourier coefficients for the number density. From this result one can also find the Fourier coefficients for the charge density, $\widetilde{\rho}_q$. First, it is observed that due to symmetry the number density follows a sine series, i.e.,

$$n_i(z) = n_0 + \sum_{j=n}^{\infty} \widetilde{n}_{i,j}(k_j)\sin(k_j z). \tag{11}$$

The Fourier components of the charge density is then

$$\widetilde{\rho}_q(k_n) = q_+\widetilde{n}_+ + q_-\widetilde{n}_- = -\left(\frac{\widetilde{\chi}_+(k_n)}{\widetilde{D}_+(k_n)} + \frac{\widetilde{\chi}_-(k_n)}{\widetilde{D}_-(k_n)}\right)\widetilde{\phi}^{\text{ext}}(k_n). \tag{12}$$

For small field strengths and negligible screening only the fundamental mode $k_n = 2\pi n/L$ is excited and we have that

$$n_i(z) \approx n_0 + \widetilde{n}_i(k_n)\sin(k_n z). \tag{13}$$

In the following, focus is on the case where Eq. (13) is true and where the two ionic species, $+$ and $-$, have same transport properties $\chi_i = \chi$, and $D_i = D$. Then Eq. (12) reduces to

$$\widetilde{\rho}_q(k_n) = -\frac{2\widetilde{\chi}(k_n)}{\widetilde{D}(k_n)}\widetilde{\phi}^{\text{ext}}(k_n). \tag{14}$$

From linear response theory [2] the Fourier components for the charge density is related to the charge-charge correlation function (or charge-charge structure) $S_{ZZ}$ by

$$\widetilde{\rho}_q(k) = -\frac{nS_{ZZ}(k)}{k_B T}\widetilde{\phi}^{\text{ext}}(k), \tag{15}$$

where $n = n_+ + n_-$. We then have an expression for $\widetilde{\chi}$ in terms of the diffusion coefficient and the charge-charge structure

$$\widetilde{\chi}(k_n) = \frac{n\widetilde{D}(k_n)}{2k_B T}S_{ZZ}(k_n). \tag{16}$$

The charge-charge structure is a collective property, and from Eq. (16) one can see that $\chi$ relates this collective property to the single particle property governed by the diffusion coefficient.

It is worth noting that in the Debye-Hückel regime, $k_B T \gg q\phi$, the charge-charge correlations are negligible, i.e., $S_{ZZ}(k) = 1$. This corresponds to the limit of zero screening and a relative permittivity of unity. Equation (16) then reads

$$\widetilde{\chi}(k) = \frac{n\widetilde{D}(k)}{2k_B T}, \quad \text{(Debye-Hückel regime)} \tag{17}$$

which is equivalent to the Nernst-Einstein equation [3] and $\chi$ can in this limit be interpreted as the ionic electric conductivity. The charge density Fourier components are in this limit $\widetilde{\rho}_q = -n\widetilde{\phi}^{\text{ext}}/k_B T$, i.e., they only dependent on amplitude of the external field. Furthermore, for systems where the diffusion coefficient is wavevector independent, $\widetilde{D}(k) \approx D_0$, the response function is

$$\widetilde{\chi}(k) = \frac{nD_0}{2k_B T}S_{zz}(k). \quad \text{(Screening regime)} \tag{18}$$

This means that the wavevector dependent response in the presence of an external electric field is dominated by the screening effects.

# 3 Simulations and results

## 3.1 Simulation details

The response is evaluated for two simple models: (i) one model for molten salt proposed by Hansen and McDonald [1] and (ii) one modified model for ionic liquids used by Chapela et al. [15]. For the molten salt the ions are simple spherical particles with same mass and point charges $\pm q$. The van der Waals interaction is the inverse power law function $V(r) = \epsilon(\sigma/r)^9$, where $r$ is the distance between two ions, $\epsilon$ and $\sigma$ define the energy and length scale, respectively. The Coulomb interactions are calculated through the shifted force method [16, 17], $\mathbf{F}(r) = q_i q_j(1/r^2 - 1/r_c^2)\mathbf{r}/r$, for $r \leq r_c$. Here $\mathbf{r}$ is the vector of separation with magnitude $r$, and $r_c$ is the cut-off radius set to $r_c = 3\sigma$; this cut-off distance is also used for the van der Waals interactions. The positions of the particles are integrated forward in time with the leap-frog algorithm [18] and the temperature is controlled using a Nosé-Hoover thermostat [19,20]. In all simulations the total ion number density is $n = 0.368\sigma^{-3}$; the number of ions are 1000, giving 500 ion-pairs. Two different temperatures are simulated, $T = 0.0177\epsilon/k_B$ and $1.0177\epsilon/k_B$, the former being a realistic temperature for the model. To simulate the Debye-Hückel regime $k_B T \gg q\phi$ the ion-ion Coulomb interactions are removed whilst keeping the temperature fixed at $T = 1.0177\epsilon/k_B$; this system is symbolized using $T_\infty$. Alternatively, one can perform simulations at very high temperatures, but this will result in numerical instabilities. In the following all quantities are given in units of $\sigma$, $\epsilon$, $q$, and mass $m$, and as it is common practise these the units are not written explicitly.

For the simple molten salt system the shifted force method can be tested against the direct Ewald summation method [21]. From equilibrium simulations it was found from the structure that the Ewald method converges satisfactory using 124 replica systems and that it agrees with the data from the shifted force method, see also Ref. [17]. For the non-equilibrium situation at $T = 1.0177$ the Ewald and shifted force methods yield same results for all wavevectors tested $0 < k < 2.2$.

The modified ionic liquid model is composed of cations with a spherical point charge particle (head group) and two spherical non-charged tail particles. The particles in the cation are linearly connected using a simple spring force $\mathbf{F} = -k(r-1)\mathbf{r}/r$, where $k = 100$ is the spring constant. Anions are simple spherical point charge particles [15]. Rather than a hard-sphere type potential in the original model, the van der Waals interactions are here given through the

Weeks-Chandler-Andersen potential [22] $V(r) = 4((1/r)^{12} - (1/r)^6)$, where the cut-off is set at $r_c = 2^{1/6}$. The Coulomb interaction is given by the Yukawa potential $V(r) = q^2 e^{-r/\lambda_D}/r$, with $\lambda_D = 1/2$ corresponding to a relative small Debye screening length and the reduced charge is $q = 4$. The cut-off distance for the Yukawa potential is set to $r_c = 2.5$. The state point is $(n, T) = (1, 1)$ and the simulation method is the same as for the molten salt simulations. This choice of parameters gives, qualitatively, the fluid structure observed in different ionic liquid [23, 24]. The number of particles are 864, that is, 216 ion pairs.

Simulations of the non-equilibrium system is also performed. Here an additional force from the external field, Eq. (3), is added to the total force experienced by the ions $\mathbf{F}_i^{ext} = q_i E^{ext} \mathbf{k}$, where $\mathbf{k}$ is the unit vector parallel to the $z$-axis.

## 3.2 Results: molten salt

The wavevector dependent diffusivity can be obtained as follows. The Gaussian approximation [2, 25] relates the diffusion coefficient to the incoherent intermediate scattering function (or the self-part of the density-density correlations), so in the diffusive regime, i.e., for large $t$, this is here generalized to

$$F_s(k, t) = e^{-\widetilde{D}(k)k^2 t}. \tag{19}$$

The Fourier-Laplace transformation is

$$S_s(k, \omega) = \int_0^\infty e^{-i\omega t} e^{-\widetilde{D}(k)k^2 t} \, \mathrm{d}t = \frac{1}{i\omega + \widetilde{D}(k)k^2}, \tag{20}$$

which gives an expression for the wavevector dependent diffusivity in the limit of zero frequency

$$\widetilde{D}(k) = \frac{1}{k^2 S_s(k, 0)}. \tag{21}$$

Microscopically the intermediate scattering function is defined from the ensemble average [25]

$$F_s(k, t) = \frac{1}{N} \left\langle \sum_i e^{-ik(z_i(t) - z_i(0))} \right\rangle, \tag{22}$$

where $N$ is the number of ions and is thus a single particle property. In Fig. 1 (a) the intermediate scattering function is plotted for different wavevectors in the case of $T = 0.0177$. Also, shown as punctured lines $f(k, t) = e^{-\frac{1}{6}\langle \Delta r^2 \rangle k^2 t}$, where $\langle \Delta r^2 \rangle$ is the particle mean square displacement. It is seen that the Gaussian approximation holds surprisingly well for this model validating Eq.(19). The data are Fourier-Laplace transformed and the Gaussian diffusion kernel is found from Eq. (21); the results are plotted in Fig. 1 (b). The function

$$\widetilde{D}(k) = D_0/(1 + \alpha k^\beta) \tag{23}$$

is fitted to data where the zero wavevector diffusion coefficient, $D_0$, is found from the mean square displacement $\langle \Delta r^2 \rangle = 2D_0 t$ for $t \to \infty$. It is observed that the normalized kernel is identical for the two cases $T = 1.0177$ and $T = T_\infty$. For $T = 0.0177$ the diffusivity features a relative low wavevector dependency in the range studied here and we have $\widetilde{\chi}(k) \propto S_{ZZ}(k)$ according to Eq. (18).

Next the charge-charge structure is evaluated. This is defined as [2]

$$S_{ZZ}(\mathbf{k}) = \frac{1}{N} \left\langle \rho_q(\mathbf{k}, 0)\rho_q(-\mathbf{k}, 0) \right\rangle, \tag{24}$$

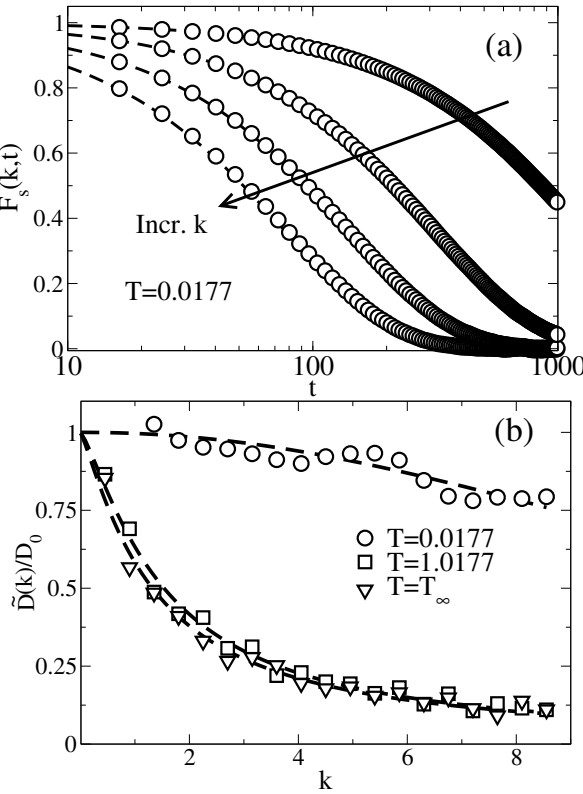

Figure 1: Molten salt (a): Incoherent intermediate scattering function for different wavevectors (circles). Punctured line is $f(k,t) = e^{-\frac{1}{6}\langle \Delta r^2 \rangle k^2 t}$, where $\langle \Delta r^2 \rangle$ is the mean square displacement. (b) The diffusion kernel at different temperatures. Punctured lines are best fit to Eq.(23). Parameter values are for $T = 0.0177, 1.0177$ and $T_\infty$, respectively: $D_0 = 0.011, 0.84, 0.92$, $\alpha = 0.0073, 0.69, 0.78$, and $\beta = 1.72, 1.14, 1.12$.

where $\rho_q(\mathbf{k}, 0) = \sum_i q_i e^{-i\mathbf{k}\cdot\mathbf{r}_i}$. Note, this is a collective property. For non-zero wavevectors $S_{zz}(k)$ can also be calculated from the radial distribution functions, see e.g. Ref. [1],

$$S_{zz}(k) = 1 + \frac{2\pi n}{k} \int_0^\infty \Delta g(r) r \sin(kr) \, dr \,, \tag{25}$$

where $\Delta g(r)$ is the difference between the cation-cation and cation-anion radial distribution functions, $\Delta g(r) = g_{++}(r) - g_{+-}(r)$. The charge-charge structure is plotted in Fig. 2 for the three different systems; symbols are data from Eq. (24) and lines are $S_{zz}$ calculated from Eq. (25). As expected we observe a zero screening, $S_{zz} = 1$, for $T = T_\infty$, but non-negligible screening for $T = 1.0177$ and $T = 0.0177$.

From Eq. (16) the Fourier components of $\chi$ can be evaluated, the results is shown in Fig. 3 (a). The evaluation is based on the fit of the diffusion kernel, Eq. (23), and the integral expression for the charge-charge structure, Eq. (25). First, for zero screening, $T = T_\infty$, the response function is monotonically decaying with respect to wavevector. This behavior is typically observed for the diffusion and viscosity kernels [8]. For non-zero screening the response features a maximum depending on temperature; the characteristic wave length $l = 2\pi/k_{max}$ where $k_{max}$ is the wavevector corresponding to maximum in $\widetilde{\chi}$, is approximately $l = 2.5$ for $T = 1.0177$ and $l = 1.6$ for $T = 0.0177$. This means that application of an external field will result in a relatively small flux, $j_i^e$, on large length scales and a maximum for wavelength of roughly 2 atomic diameters. For $T = 0.0177$, we have that $\lim_{k\to 0} \widetilde{\chi}(k) = 0$ which means that the charge density is zero at these length scales; this is in agreement with perfect screening.

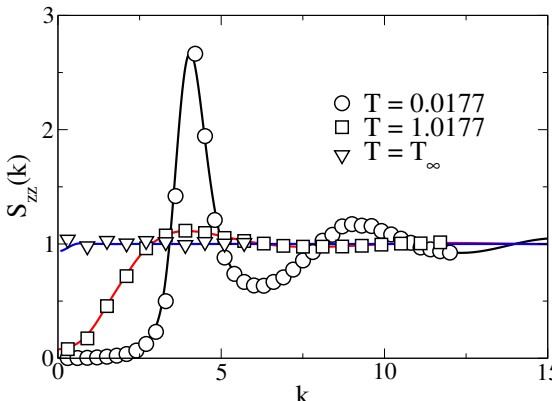

Figure 2: Charge-Charge structure for the molten salt model. Symbols are data obtained from Eq. (24) and full lines are from Eq. (25)

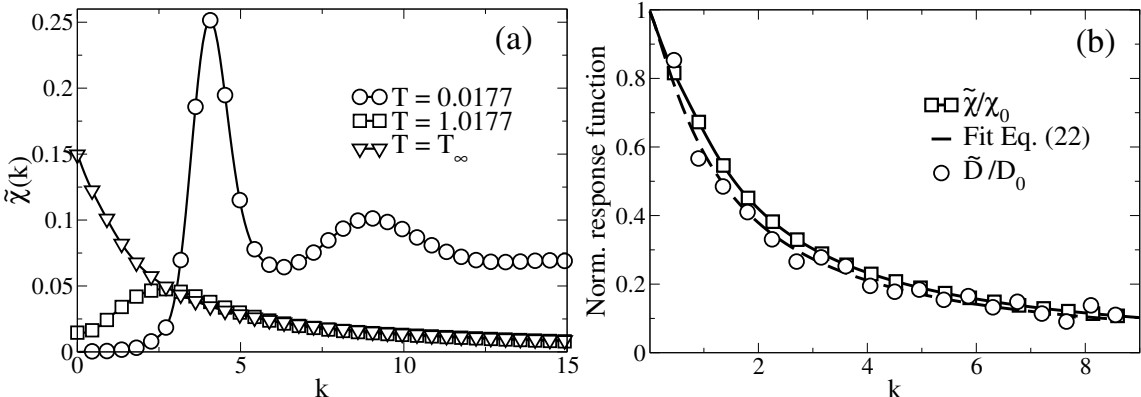

Figure 3: Molten salt (a) The Fourier components of $\chi$. (b) Normalized kernels, $\widetilde{\chi}/\chi_0$ and $\widetilde{D}/D_0$, for $T = T_\infty$.

Another important point is that $\lim_{k\to\infty} S_{zz} = 1$, and if $\lim_{k\to\infty} D = 0$ as indicated in Fig. 1 we have that $\lim_{k\to\infty} \widetilde{\chi} = 0$ according to Eq. (16).

In Fig. 3 (b) $\widetilde{\chi}(k)/\chi_0$ and $\widetilde{D}(k)/D_0$ are depicted for the case $T = T_\infty$. The data show good collapse, that is, there exists a master curve response function. This identical wavevector dependence indicates that the response functions are governed by the same underlying process. Specifically, it is here conjectured that the $\chi$-response is given by the diffusion processes in the system, i.e., cross correlation effects can be ignored. From Fig. 3 (a) one can immediately see that this collapse is not found for the $T = 1.0177$ and $T = 0.0177$ cases, hence, different processes are involved.

The theory is compared with the non-equilibrium simulations. Figure 4 (a) shows the charge density profile, $\rho_q$, for two wavevectors $k = 2\pi/L$ and $k = 8\pi/L$ at $T = 0.0177$. The system length is $L = 13.955$ and $m = 1$, hence, the potential field amplitude is constant. It is observed that the charge density amplitude is larger for smaller wavelengths as expected. For $k > 12\pi/L$ a simple spectral analysis shows that higher order modes are excited compromising Eq. (13) and only results for $k < 12\pi/L$ is shown. Figure 4 (b) compares the amplitude for all three temperatures with the predictions from the theory, Eq. (14). The agreement is excellent. Of course, this comparison is equivalent to test the linear response, Eq.(15). The case of $m = 0$ is also shown, however, the agreement is less satisfactory for low wavevectors, which is due to the diverging amplitude in the limit of zero wavevector causing a non-linear response and failure of the constitutive relation, Eq. (7).

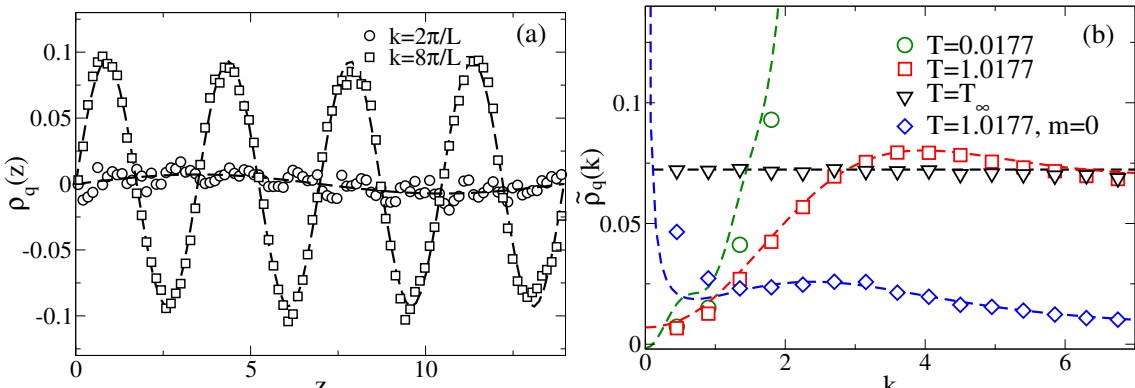

Figure 4: Non-equilibrium results for molten salt (a) Charge density profiles for $T = 0.0177$. Lines are sine functions with amplitudes $\widetilde{\rho}_q = 0.0072$ and $0.093$, values obtained from a spectral analysis. (b) Charge density amplitudes for all three temperatures and for $m = 0$. Symbols are simulation results. Lines are predictions from the theory, Eq. (14).

### 3.3 Results: ionic liquid

In Fig. 5 (a) the diffusion kernels are is shown for the ion liquid model. These are evaluated as explained in Sect. 3.2. One sees that within statistical uncertainty the diffusion kernel is wavevector independent, at least up to $k = 5$. Beyond this wavevector value the statistical error increases dramatically and the results are non-conclusive. The charge-charge structure, Fig. 5 (b), is calculated from the direct definition Eq. (24). It features relatively strong structure, that is, the system is in the screening regime. We can therefore expect the response function $\chi$ to resemble low temperature molten salt response function.

For the ionic liquid Eqs. (13)-(18) do not apply as $\chi_+ \neq \chi_-$ and $D_+ \neq D_-$, and $\widetilde{\chi}_i$ is found from non-equilibrium simulations using Eq. (10) directly. This also means that we cannot compare the predictions from these equations with simulation data. The amplitudes of the density profiles for both the anion and cation are analyzed giving $\widetilde{n}_i$. Note that only single modes are excited for the low external field applied, $E_0 = 0.05$. Substitution of $\widetilde{n}_i$ and $\widetilde{D} = D_0$ into Eq. (10) yields the results in Fig. 5 (c). The response features a maximum for $k \approx 4.25$ in good agreement with the maximum charge-charge structure.

To investigate if the two kernels can be mapped onto the same master curve, the results from the cation kernel is normalized with respect the maximum. The normalized result is shown in Fig. 5 (c) as squares connected with a punctured line. To a reasonable agreement the two kernels do follow a master curve which indicates that the underlying mechanisms responsible for the response are the same. This contrasts the wavevector independent diffusion kernel, that is, the system response seen in the mass flux due to the density gradient. Therefore, the physical mechanisms for the two fluxes $j_i^d$ and $j_i^e$ are fundamentally different; at least in the screening regime.

## 4 Conclusion

In this paper the mass flux of an ionic system due to a spatially varying electric field is studied. Following the Nernst-Planck equation two forces are present in this system: (i) the concentration gradient and (ii) the gradient of the electric potential. The two response functions (or kernels) that account for the system response to these forces are the diffusion- and $\chi$-response functions; the $\chi$-response function relates the mass flux with the external electric

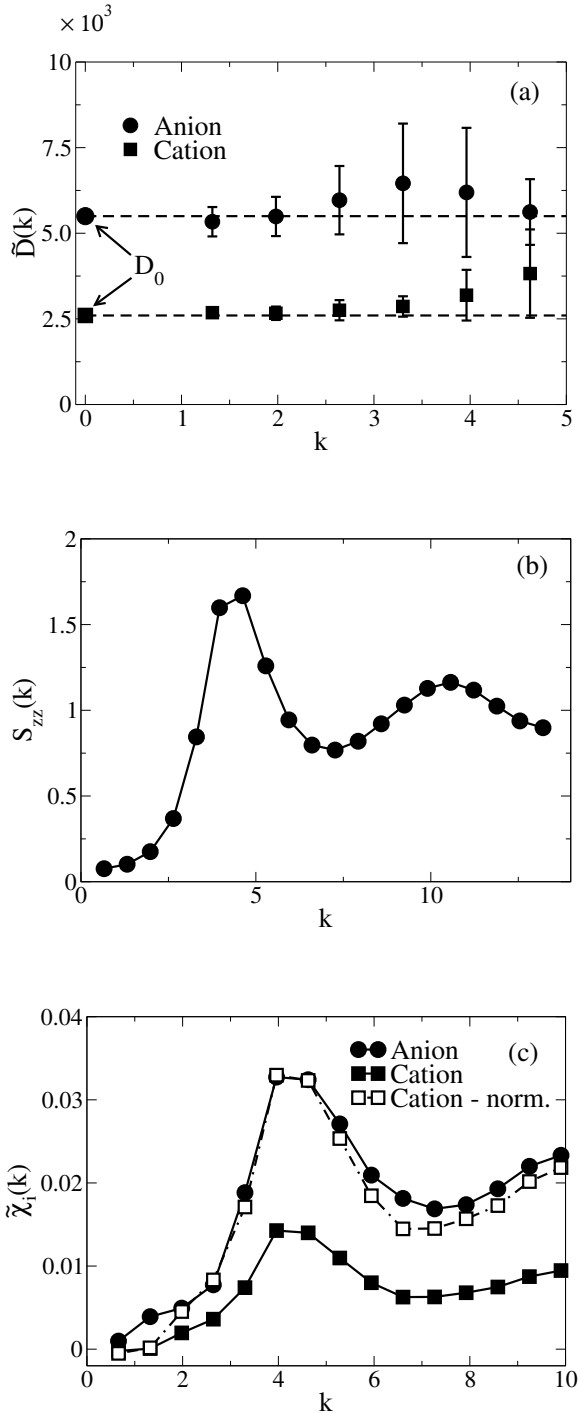

Figure 5: Ionic liquid. (a) Diffusion kernels for the anion and cation. (b) Charge-charge structure. (c) wavevector dependent response function $\widetilde{\chi}_i$ for the anion and cation. Squares connected with punctured line is the normalized cation response function. For all figures lines serve as a guide to the eye.

field excluding the contribution from the concentration gradient (here modelled through the self-diffusion). Note, this differs from the charge-charge response function, $S_{zz}$, which relates the charge density to the electric field including all underlying processes, and the ionic conductivity that relates the charge current to the local field. In the limit of zero screening the $\chi$-response function is directly related to the conductivity, on the other hand, in the large

screening regime the response function is related to the charge-charge structure.

The spatial correlations in the system are manifested in the wavevector dependence of the kernels. The molecular dynamics simulation data show the diffusion and $\chi$-kernels feature very different wavevector dependence in the screening regime. Interestingly, in the screening regime both the molten salt and ionic liquid feature a wavevector independent diffusion kernel and the response to the external field is dominated by the charge-charge structure. This latter quantity is a collective property of the system. In the non-screening regime, on the other hand, the response to the external field is closely related to the ionic conductivity and in this regime the Nernst-Einstein relation holds to a good approximation, i.e., cross-correlation effects are negligible.

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
