# Peer review of "Multiscale response of ionic systems to a spatially varying electric field"

_SciPost Physics, doi:SciPost Phys. 2, 017 (2017)_

## Round 2 · Referee Report · Anonymous · 2016-12-15

Strengths

1. The calculations seem to be correct in theoretical part.

Weaknesses

1. The situation is unrealistic.
2. The molecular simulation has no validation, and then the agreements between the theoretical results and the numerical results are not acceptable (at least for me).

Report

1. The authour calculates the response of ions under spatially varying (but stationary) electric field, as shown in Eq.(3). However, as he admits,
''The experimental realization of this field is not straightforward''. This point greatly decrease the significance of the present manuscript. The authour should propose a possible experimental realisation of this ''unrealistic'' electric field.

2. In numerical simulation, the authour uses the shifted Coulomb force, introducing some references. However, I can never believe the validity of this approximation, because the long-range nature of Coulomb interaction is disregarded in this method. The authour should compare his results with other methods treating the Coulomb interaction sincerely, e.g. the Ewald method.

Requested changes

Minor comment:
1. In molecular simulations, the numerical unit of the charge is given by $\sqrt{\epsilon\sigma}$. How the authour relate this and $q$?

2. In page 7, ''Yurukawa'' should be corrected as ''Yukawa''. In the caption of Fig.1, text overlap of ''lines'' in the third line.

  • validity: low
  • significance: low
  • originality: ok
  • clarity: high
  • formatting: excellent
  • grammar: excellent

Author:  Jesper Schmidt Hansen  on 2017-02-15  [id 100]

(in reply to Report 1 on 2016-12-15)
Category:
answer to question

I have attached my reply to the referee comments.

Attachment:

reply-ref2.pdf

---

## Round 2 · Referee Report · Anonymous · 2016-12-22

Strengths

1. The development of simple relations between dynamic properties in the presence of an external field and equilibrium properties of ionic systems is an interesting and important problem.

2. The authors proceeds to compute inputs to the theory from simulation.

Weaknesses

1. Many issues that need to be clarified regarding the physical interpretation of the results (see report).

2. There are issues with the simulation procedure. In particular, is the use of cutoffs reasonable for all systems studied.

3. The ionic liquid model needs to be described in much greater detail.

4. The author never actually uses the theory to describe the response of fluids to external fields. Even though this is the purpose of the theory.

Report

\newcommand{\kT}{\ensuremath{k_{\rm B}T}}

%%
%
In this paper, Hansen investigates the response of ionic systems to a spatially varying electric field through
a combination of theory and molecular dynamics simulations.
%
\textbf{This work may be suitable for publication eventually, but only after significant revision.}
%
There are some shortcomings that need to be addressed, as detailed below.
%
However, addressing these issues should significantly improve the paper.
%

\begin{itemize}
\item After Eq. (5), the author assumes that the response function $\chi_N'$ can be separated into distinct spatial
and temporal components. Can the author explain why this assumption is valid and when it is expected to break down?

\item The sentence ``The symbol $\chi_i$ is a matter of choice." is not necessary, because all notation is a matter of choice.

\item The author refers to the Debye-H\"{u}ckel regime as the ``zero screening'' limit throughout.
The meaning behind this statement would be better appreciated by the reader if the author mentioned the dielectric constant, $\varepsilon$.
For example, Eq. (17) implies
\begin{equation}
\frac{1}{\varepsilon(k)} = 1+\frac{2\pi n \tilde{D}(k)}{\kT k^2},
\end{equation}
such that $\varepsilon(k)\rightarrow 0$ as $k\rightarrow 0$,
assuming $\tilde{D}(k)$ does not impact the small $k$-dependence of $\varepsilon$.
Note that $\varepsilon(0)=\infty$ in the complete or perfect screening limit.
See Chapter 10 of Reference 2 for further details.

\item Similarly, in the ``screening regime,'' the dielectric constant is (using Eq. 18)
\begin{equation}
\frac{1}{\varepsilon(k)} = 1+\frac{2\pi n D_0}{\kT k^2} S_{zz}(k).
\end{equation}
Using $\lim_{k\rightarrow 0} S_{zz}(k) = q^2 k^2 / [k_{\rm D}^2 (1+k^2/k_s^2)]$, where $k_{\rm D}$ is the Debye wavelength,
$k_s=A k_{\rm D}^2$, and $A$ is related to the isothermal compressibility (See Chapter 10 of Ref. 2),
one can readily obtain $\lim_{k\rightarrow 0} \varepsilon^{-1}(k)=1+D_0/2$.
Thus, in the perfect screening regime, $\varepsilon(k)=\infty$ implies $D_0=-2$.

Therefore, when complete screening is obtained, $D_0$ is negative! Is there a sign error somewhere in the text?
Or is $\tilde{D}(k)<0$?
It is strange that Equation (10) of the manuscript (and those that follow from it) show that the nonuniform
density $n$ is proportional to the negative of $\chi$; typically $n\propto \chi$ and $n\propto - S_{zz}$.
Similarly, Eq. 18 shows that $\chi\propto S_{zz}$, while the linear response function is typically
proportional to the negative of $S_{\rm zz}$. Can the author provide some insight here?

\item On page 6, ``$\tilde{\rho}_q=n \tilde{\phi}^{\rm ext} / \kT$'' is missing a negative sign, and should read
``$\tilde{\rho}_q=-n \tilde{\phi}^{\rm ext} / \kT$''.

\item I have serious concerns with the simulation procedure.
The author uses the same fixed cutoff radius to evaluate the Coulomb interactions in the various systems.
However, unless this cutoff is significantly larger than the screening length in the systems studied,
artifacts will be introduced. Such artifacts may substantially alter the expected behavior in the system,
especially with respect to subtleties like electroneutrality that are important for examining scaling relations.

What are the screening lengths of the various systems, and how do the results change with different cutoffs?
Cutoff procedures may be a good approximation for some aspects of uniform systems, but typically break
down in nonuniform systems due to the importance of long-ranged interactions (like systems in the presence of an external field).
Why not use Ewald summation?

\item The author ``simulates'' the Debye-H\"{u}ckel regime by simply removing the charges on the ions.
Why not just simulate at a high temperature?

In this case, the author is simply studying a WCA fluid at the prescribed temperature and pressure.
Such a system has no electrostatic correlations, and the resulting charge correlations are trivial
(they are just the WCA number density correlations).

\item ``Yurukawa potential'' should be ``\textbf{Yukawa} potential'' throughout.

\item The ionic liquid model cited by the author (Ref. 15) is not what is being used in this work.
That model consists of hard sphere and square well potentials, in addition to Yukawa potentials.
Ref. 15 discretizes the Yukawa potential to simulate this model with discrete molecular dynamics simulations.

In contrast, Hansen has modified the model to consist of WCA potentials and a continuous Yukawa potential.
The author needs to describe the model in much greater detail.
What are the WCA parameters for each site? How were they chosen? What are the charges, units, etc.?

\item With regard to the fits using Eq. 23, what are the values of $D_0$, $\alpha$, and $\beta$ that are obtained?

\item At the end of page 8, Hansen states ``$\tilde{\chi}(k)\propto S_{\rm ZZ}(k)$.''
The typical charge-charge linear response function for the response of an ionic fluid to a general external field
is equal to $-\frac{n}{\kT} S_{\rm ZZ}(k)$, where $n$ is the bulk density of the fluid.
How does $\tilde{\chi}(k)$ relate to the usual charge-charge linear response function,
and in particular, how does its physical interpretation differ?

\item Do the charge-charge structure factors satisfy the zeroth and second moment conditions of Stillinger and Lovett?

\item How are the values of $l$ obtained on page 10?

\item The discussion of the ``$T_\infty$'' model on page 10 is trivial (it is an uncharged, WCA fluid).
I would argue that the results for the $T=1.0177$
and $T=0.0177$ systems should be more interesting. This data should be included and discussed.

\item Page 10, last paragraph, ``exited'' should be ``excited''

\item Fig. 4, what about larger $k$-values for the $T=0.0177$ curve? What does this data look like?

\item For all the simulations in the presence of an external field, the author should compare the results
of the theory in Section II with the simulation results.
Without this comparison, there is little reason to develop the theory at all.
The author should also do this for the ionic liquid section.

\end{itemize}

Requested changes

1. Clarify assumptions in the theory.

2. Clarify language regarding "zero screening" and "screening" regimes.

3. Correct typographical errors.

4. Verify the correctness of the simulation procedure.

5. Give more details regarding the simulations.

6. Discuss in more detail the physical meaning of the response function $\chi$ and how it relates to the typical charge-charge linear response function.

7. Compare the theory developed to the simulation results in the presence of an external field.

  • validity: good
  • significance: good
  • originality: ok
  • clarity: low
  • formatting: good
  • grammar: excellent

Author:  Jesper Schmidt Hansen  on 2017-02-15  [id 99]

(in reply to Report 2 on 2016-12-22)
Category:
answer to question

I have attached my reply to the referee.

Attachment:

reply-ref1.pdf

---

## Round 3 · Referee Report · Anonymous (Referee 2) · 2017-3-22

Strengths

  1. The development of simple relations between dynamic properties in the presence of an external field and equilibrium properties of ionic systems is an interesting and important problem.

  2. The authors proceeds to compute inputs to the theory from simulation.

  3. Theory accurately reproduces numerical results.

  4. Creates understanding of response functions in charged systems.

Weaknesses

None

Report

The author has adequately addressed the comments and concerns of both reviewers. I feel that the paper is now significantly improved and should be published.

Requested changes

None

---

## Round 3 · Referee Report · Anonymous (Referee 1) · 2017-3-24

Strengths

already addressed in the previous report

Weaknesses

already addressed in the previous report

Report

I do not change my opinion about ``experimental realisation'' and ``the validity of the shifted Coulomb force method'', but the authour well reply to my questions and I can suggest the manuscript for publication.

Requested changes

none

---

## Round 3 · Author Response

Dear Editor

I hereby submit the revised version of the manuscript "Multiscale response of ionic systems to a spatially varying electric field".

Best regards Jesper

---

## Round 3 · List of Changes

The changes are highlighted in red. Also, please refer to the reply to the referees.

---

## Editorial Decision

published